# Ameliorative Effects of a Rhenium (V) Compound with Uracil-Derived Ligand Markers Associated with Hyperglycaemia-Induced Renal Dysfunction in Diet-Induced Prediabetic Rats

**DOI:** 10.3390/ijms232315400

**Published:** 2022-12-06

**Authors:** Angezwa Siboto, Akinjide Moses Akinnuga, Bongiwe Khumalo, Muhammed Bilaal Ismail, Irvin Noel Booysen, Ntethelelo Hopewell Sibiya, Phikelelani Ngubane, Andile Khathi

**Affiliations:** 1School of Laboratory Medicine and Medical Sciences, College of Health Sciences, University of KwaZulu-Natal, Durban 4000, South Africa; 2School of Chemistry and Physics, College of Agriculture, Engineering Sciences, University of KwaZulu-Natal, Pietermaritzburg 3209, South Africa; 3Pharmacology Division, Rhodes University, Grahamstown 6140, South Africa

**Keywords:** rhenium (V) compound, prediabetes, dietary modification, kidney dysfunction, oxidative stress

## Abstract

Kidney disease is characterised by the improper functioning of the kidney as a result of kidney damage caused by hyperglycaemia-induced oxidative stress. The moderate hyperglycaemia seen in prediabetes can be treated using a combination of metformin and lifestyle interventions (low-calorie diets and exercising). However, patients have been reported to over-rely on pharmacological interventions, thus decreasing the efficacy of metformin, which leads to the development of type 2 diabetes mellitus (T2DM). In this study, we investigated the effects of a rhenium (V) compound in ameliorating renal dysfunction in both the presence and absence of dietary modification. Kidney function parameters, such as fluid intake and urine output, glomerular filtration rate (GFR), kidney injury molecule (KIM 1), creatinine, urea, albumin and electrolytes, were measured after 12 weeks of treatment. After treatment with the rhenium (V) compound, kidney function was restored, as evidenced by increased GRF and reduced KIM 1, podocin and aldosterone. The rhenium (V) compound ameliorated kidney function by preventing hyperglycaemia-induced oxidative stress in the kidney in both the presence and absence of dietary modification.

## 1. Introduction

The prevalence of type 2 diabetes mellitus (T2DM) is increasing in African countries due to population growth, urbanisation and sedentary lifestyles [1,2]. Epidemiological studies indicate that 65–75% of the risk of primary hypertension is due to obesity and prediabetes [1,3]. At least 72% of patients with end-stage renal disease have hypertension and T2DM, both driven largely by obesity [2,3]. Kidney dysfunction as a consequence of T2DM might be reinforced by the presence of other diabetic complications, such as cardiovascular disease and hypertension [3,4]. Kidney dysfunction is associated with albuminuria, a lowered glomerular filtration rate, glomerulosclerosis, electrolyte abnormalities and proteinuria [5,6]. Other factors such as hyperglycaemia, increased oxidative stress, chronic inflammation, impaired insulin signalling, dyslipidaemia, renal polyol formation and the accumulation of advanced glycation end-products can also significantly contribute to the onset and progression of kidney disease [7,8].

Prediabetes is a progressive disorder that often precedes the onset of T2DM, which is characterised by hyperglycaemia and hyperinsulinaemia [9,10]. Hyperglycaemia-induced oxidative stress leads to kidney damage as a result of the harmful effects of the over-production of reactive oxygen species (ROS) in the mitochondria [11,12]. Kidney disease is characterised by an increase in kidney size, the improper functioning of the kidney, an increased level of albumin in the urine, glomerular basement membrane (GBM) thickening and glomerular hypertrophy [7,13]. Therefore, there is a need for treatments that can ameliorate the hyperglycaemia seen in prediabetes to prevent the onset of kidney disease [14].

Metformin has been used to manage prediabetes and its associated complications [15,16]. However, the use of metformin is more effective when used in conjunction with lifestyle interventions [17]. These include the consumption of low-sodium, low-calorie diets and increased physical activity [16,17]. There have been reports of the under-utilisation of lifestyle interventions and an over-reliance on the use of metformin, leading to poor clinical outcomes; therefore, there is a need for alternative pharmacological treatments that are effective with and without lifestyle interventions [14,15,17]. A previous study in our laboratory showed that a rhenium (V) compound is able to reduce blood glucose by increasing the expression of GLUT 4 in peripheral tissues, such as skeletal muscle [18,19]. Furthermore, the rhenium (V) compound ameliorated liver injury and prevented hepatotoxicity in prediabetic rats in both the absence and presence of a diet intervention [20].

The rhenium (V) compound is a metal-based compound that has a biologically active ligand system [21]. A study conducted by Maisuls et al. (2017) reported that rhenium (V) compounds allow for a great deal of structural and chemical variability and can be fine-tuned to meet the requirements of a wide range of biological applications (fluorescence markers and/or antitumor drugs) [22,23]. The ligands provide stability and promote the bio-availability of the metal complex [21,22]. This study sought to investigate whether a rhenium (V) compound with uracil-derived ligands can ameliorate renal dysfunction associated with prediabetes in both the absence and presence of a diet intervention in diet-induced prediabetic rats.

## 2. Results

### 2.1. Effects of Rhenium (V) Compound Administration in the Absence and Presence of Diet Intervention on Glucose, Insulin Levels and HOMA2-IR Index

Figure 1 shows the glucose levels in the normal control (NC), prediabetic control (PD), metformin and diet intervention (MET + DI), metformin and high-fat/high-carbohydrate diet (MET + HFHC), rhenium (V) compound and diet intervention (Re + DI) and rhenium (V) compound and high-fat/high-carbohydrate diet (Re + HFHC) groups after 12 weeks of treatment. In a comparative evaluation with the NC group, PD showed a significant increase in glucose levels (*p* < 0.05; Figure 1), and in comparison with the PD control group, the Re + DI and Re + HFHC groups showed a significant decrease in glucose levels (*p* < 0.05). See Figure 1.

Figure 2A,B show the insulin concentration and HOMA2-IR index in the normal control (NC), prediabetic control (PD), metformin and diet intervention (MET + DI), metformin and high-fat/high-carbohydrate diet (MET + HFHC), rhenium (V) compound and diet intervention (Re + DI) and rhenium (V) compound and high-fat/high-carbohydrate (Re + HFHC) groups after 12 weeks of treatment. In a comparative evaluation with the NC group, PD showed a significant increase in insulin concentration and HOMA2-IR (*p* < 0.05; Figure 2A,B), and in comparison with the PD control group, the Re + DI and Re +HFHC groups showed a significant decrease in insulin concentration and the HOMA2-IR index (*p* < 0.05). See Figure 2A,B.

### 2.2. Effects of Rhenium (V) Compound Administration in the Absence and Presence of Diet Intervention on Renal Oxidative Stress and Antioxidant Status

Figure 3, Figure 4 and Figure 5 show the lipid peroxidation and antioxidant enzyme activities (SOD and GPx) in the normal control (NC), prediabetic control (PD), metformin and diet intervention (MET + DI), metformin and high-fat/high-carbohydrate diet (MET + HFHC), rhenium (V) compound and diet intervention (Re + DI) and rhenium (V) compound and high-fat/high-carbohydrate diet (Re + HFHC) groups after 12 weeks of treatment. For lipid peroxidation (Figure 3), in comparison with the NC group, PD showed a significant increase in MDA concentration in kidney tissue (*p* < 0.05; Figure 3). In comparison with the PD control group, the Re + DI and Re + HFHC groups showed a significant decrease in MDA concentration (*p* < 0.05). A similar effect was observed in the MET + DI group, with a significantly decreased MDA concentration (*p* < 0.05). For antioxidant activity (Figure 4 and Figure 5), in comparison with the NC group, PD showed a significant decrease in both GPx and SOD activities in kidney tissue (*p* < 0.05; Figure 4 and Figure 5). In comparison with the PD control group, the Re + DI and Re + HFHC groups showed a significant increase in both GPx and SOD activities (*p* < 0.05). A similar effect was observed in the MET + DI group, with a significant increase in antioxidant enzyme activities (*p* < 0.05). See Figure 4 and Figure 5.

### 2.3. Effects of Rhenium (V) Compound Administration in the Absence and Presence of Diet Intervention on Inflammatory Markers: TNF-α and IL-6

Figure 6 shows the TNF-α levels in the normal control (NC), prediabetic control (PD), metformin and diet intervention (MET + DI), metformin and high-fat/high-carbohydrate diet (MET + HFHC), rhenium (V) compound and diet intervention (Re + DI) and rhenium (V) compound and high-fat/high-carbohydrate diet (Re + HFHC) groups after 12 weeks of treatment. In a comparative evaluation with the NC group, PD showed a significant increase in TNF-α levels (*p* < 0.05; Figure 6), and in comparison with the PD control group, Re + DI and Re + HFHC groups showed a significant decrease in TNF-α levels (*p* < 0.05). See Figure 6.

Figure 7 shows the IL-6 levels in the normal control (NC), prediabetic control (PD), metformin and diet intervention (MET + DI), metformin and high-fat/high-carbohydrate diet (MET + HFHC), rhenium (V) compound and diet intervention (Re + DI) and rhenium (V) compound and high-fat/high-carbohydrate diet (Re + HFHC) groups after 12 weeks of treatment. In a comparative evaluation with the NC group, PD showed a significant increase in IL-6 levels (*p* < 0.05; Figure 6), and in comparison with the PD control group, Re + DI and Re +HFHC groups showed a significant decrease in IL-6 (*p* < 0.05). See Figure 7.

### 2.4. Effects of Rhenium (V) Compound Administration in the Absence and Presence of Diet Intervention on KIM 1 and GFR

Figure 8 shows the KIM-1 concentration in the normal control (NC), prediabetic control (PD), metformin and diet intervention (MET + DI), metformin and high-fat/high-carbohydrate diet (MET + HFHC), rhenium (V) compound and diet intervention (Re + DI) and rhenium (V) compound and high-fat/high-carbohydrate diet (Re + HFHC) groups after 12 weeks of treatment. In comparison with the NC group, the PD group showed a significant increase in KIM-1 concentration (*p* < 0.05) (Figure 8). The administration of the rhenium (V) compound in both the diet intervention and high-fat/high-carbohydrate treatments resulted in a significant decrease in KIM-1 concentration when compared with the PD group (*p* < 0.05). A similar effect was observed with the MET + DI-treated group when compared with the PD group (*p* < 0.05). See Figure 8.

Figure 9 shows the GFR in the normal control (NC), prediabetic control (PD), metformin and diet intervention (MET + DI), metformin and high-fat/high-carbohydrate diet (MET + HFHC), rhenium (V) compound and diet intervention (Re + DI) and rhenium (V) compound and high-fat/high-carbohydrate diet (Re + HFHC) groups after 12 weeks of treatment. In comparison with the NC group, the PD group showed a significant decrease in GFR (*p* < 0.05) (Figure 2). The administration of the rhenium (V) compound in both the diet intervention and high-fat/high-carbohydrate treatments resulted in a significant increase in GFR when compared with the PD group (*p* < 0.05). A similar effect was observed with the MET + DI-treated group when compared with the PD group (*p* < 0.05). See Figure 2.

### 2.5. Effects of Rhenium (V) Compound Administration in the Absence and Presence of Diet Intervention on Plasma and Urinary Sodium and Potassium (Electrolytes Na^+^ and K^+)^, Fluid Intake and Urine Output

Figure 10 shows the fluid intake in the normal control (NC), prediabetic control (PD), metformin and diet intervention (MET + DI), metformin and high-fat/high-carbohydrate diet (MET + HFHC), rhenium (V) compound and diet intervention (Re + DI), and rhenium (V) compound and high-fat/high-carbohydrate diet (Re + HFHC) groups after 12 weeks of treatment. There was a significant increase in fluid intake in PD when compared to NC (*p* < 0.05). After treatment for 12 weeks with the rhenium (V) compound, there was a noticeable significant decrease in fluid intake and urine output in treated groups when compared to PD (*p* < 0.05 Figure 10). A similar effect was observed with the MET + DI-treated group when compared with the PD group (*p* < 0.05). A similar effect was observed with the MET + DI-treated group when compared with the PD group. See Figure 10.

Figure 11 shows the urine output in the normal control (NC), prediabetic control (PD), metformin and diet intervention (MET + DI), metformin and high-fat/high-carbohydrate diet (MET + HFHC), rhenium (V) compound and diet intervention (Re + DI) and rhenium (V) compound and high-fat/high-carbohydrate diet (Re + HFHC) groups after 12 weeks of treatment. There was a significant increase in urine output in PD when compared to NC (*p* < 0.05). After treatment for 12 weeks with the rhenium (V) compound, there was a noticeable significant decrease in fluid intake and urine output in treated groups when compared to PD (*p* < 0.05 Figure 11). A similar effect was observed with the MET + DI-treated group when compared with the PD group (*p* < 0.05). See Figure 11.

Figure 12A,B show plasma and urinary potassium (K^+^) in the normal control (NC), prediabetic control (PD), metformin and diet intervention (MET + DI), metformin and high-fat/high-carbohydrate diet (MET + HFHC), rhenium (V) compound and diet intervention (Re + DI) and rhenium (V) compound and high-fat/high-carbohydrate diet (Re + HFHC) groups after 12 weeks of treatment. In Figure 12A, there is a significant increase in urinary k^+^ concentrations in the PD group when compared to the NC group (*p* < 0.05). In contrast, after 12 weeks of treatment with the rhenium (V) compound, there was a significant decrease in urinary K^+^ concentration when compared with the PD group. In Figure 12B, there was a significant increase in plasma K^+^ concentration in the PD group when compared to the NC group (*p* < 0.05 Figure 12B). In contrast, after 12 weeks of treatment with the rhenium (V) compound, there was a significant increase in plasma K^+^ concentration in the rhenium (V) compound-treated groups when compared with the PD group (*p* < 0.05). A similar effect was observed with the MET + DI-treated group when compared with the PD group. See Figure 12A,B.

Figure 13A,B show plasma and urinary sodium in the normal control (NC), prediabetic control (PD), metformin and diet intervention (MET + DI), metformin and high-fat/high-carbohydrate diet (MET + HFHC), rhenium (V) compound and diet intervention (Re + DI), and rhenium (V) compound and high-fat/high-carbohydrate diet (Re + HFHC) groups after 12 weeks of treatment. In Figure 13A, there is a significant decrease in urinary Na^+^ concentrations in the PD group when compared to the NC group (*p* < 0.05). In contrast, after 12 weeks of treatment with the rhenium (V) compound, there was a significant increase in urinary Na+ concentration when compared with the PD group. In Figure 13B, there is a significant increase in plasma Na^+^ concentration in the PD group when compared to the NC group (*p* < 0.05 Figure 13B). In contrast, after 12 weeks of treatment with the rhenium (V) compound, there was a significant decrease in plasma Na^+^ concentration in the rhenium (V) compound-treated groups when compared with the PD group (*p* < 0.05). A similar effect was observed with the MET + DI-treated group when compared with the PD group. See Figure 13A,B.

### 2.6. Effects of Rhenium (V) Compound Administration in the Absence and Presence of Diet Intervention on Albumin Uric Acid, Urea and Creatinine (Both Plasma and Urine)

Figure 14A,B show plasma and urinary albumin concentrations in the normal control (NC), prediabetic control (PD), metformin and diet intervention (MET + DI), metformin and high-fat/high-carbohydrate diet (MET + HFHC), rhenium (V) compound and diet intervention (Re + DI) and rhenium (V) compound and high-fat/high-carbohydrate diet (Re + HFHC) groups after 12 weeks of treatment. For plasma albumin, in comparison with the NC group, the PD group showed a significant increase in plasma albumin concentration (*p* < 0.05) (Figure 14A). The administration of the rhenium (V) compound in both the diet intervention and high-fat/high-carbohydrate treatments resulted in a significant decrease in plasma albumin concentration when compared with the PD group (*p* < 0.05). For urinary albumin, in comparison with the NC group, the PD group showed a significant decrease in urinary albumin concentration (*p* < 0.05) (Figure 14B). The administration of the rhenium (V) compound in both the diet intervention and high-fat/high-carbohydrate treatments resulted in a significant increase in urinary albumin concentration when compared with the PD group (*p* < 0.05). A similar effect was observed with the MET + DI-treated group when compared with the PD group (*p* < 0.05). See Figure 14A,B.

Figure 15A,B show plasma and urinary uric acid concentrations in the normal control (NC), prediabetic control (PD), metformin and diet intervention (MET + DI), metformin and high-fat/high-carbohydrate diet (MET + HFHC), rhenium (V) compound and diet intervention (Re + DI) and rhenium (V) compound and high-fat/high-carbohydrate diet (Re + HFHC) groups after 12 weeks of treatment. For both plasma and urinary uric acid, in comparison with the NC group, the PD group showed a significant increase in plasma and urinary uric acid concentrations (*p* < 0.05) (Figure 15A,B). The administration of the rhenium (V) compound in both the diet intervention and high-fat/high-carbohydrate treatments resulted in a significant decrease in plasma and urinary uric acid concentrations when compared with the PD group (*p* < 0.05). A similar effect was observed with the MET + DI-treated group when compared with the PD group (*p* < 0.05). See Figure 15A,B.

Figure 16A,B show urinary and plasma urea concentrations in the normal control (NC), prediabetic control (PD), metformin and diet intervention (MET + DI), metformin and high-fat/high-carbohydrate diet (MET + HFHC), rhenium (V) compound and diet intervention (Re + DI) and rhenium (V) compound and high-fat/high-carbohydrate diet (Re + HFHC) groups after 12 weeks of treatment. For plasma urea, in comparison with the NC group, the PD group showed a significant increase in plasma urea concentration (*p* < 0.05) (Figure 16A). The administration of the rhenium (V) compound in both the diet intervention and high-fat/high-carbohydrate treatments resulted in a significant decrease in plasma urea concentration when compared with the PD group (*p* < 0.05). For urinary urea, in comparison with the NC group, the PD group showed a significant decrease in urinary urea concentration (*p* < 0.05) (Figure 16B). The administration of the rhenium (V) compound in both the diet intervention and high-fat/high-carbohydrate treatments resulted in a significant increase in urinary urea concentration when compared with the PD group (*p* < 0.05). A similar effect was observed with the MET + DI-treated group when compared with the PD group (*p* < 0.05). See Figure 16A,B.

Figure 17A,B show urinary and plasma creatinine concentrations in the normal control (NC), prediabetic control (PD), metformin and diet intervention (MET + DI), metformin and high-fat/high-carbohydrate diet (MET + HFHC), rhenium (V) compound and diet intervention (Re + DI) and rhenium (V) compound and high-fat/high-carbohydrate diet (Re + HFHC) groups after 12 weeks of treatment. For plasma creatinine, in comparison with the NC group, the PD group showed a significant increase in plasma creatinine concentration (*p* < 0.05) (Figure 17A). The administration of the rhenium (V) compound in both the diet intervention and high-fat/high-carbohydrate treatments resulted in a significant decrease in plasma creatinine concentration when compared with the PD group (*p* < 0.05). For urinary creatinine, in comparison with the NC group, the PD group showed a significant decrease in urinary albumin concentration (*p* < 0.05) (Figure 17B). The administration of the rhenium (V) compound in both the diet intervention and high-fat/high-carbohydrate treatments resulted in a significant increase in urinary creatinine concentration when compared with the PD group (*p* < 0.05). A similar effect was observed with the MET + DI-treated group when compared with the PD group (*p* < 0.05). See Figure 17A,B.

### 2.7. Effects of Rhenium (V) Compound Administration in the Absence and Presence of Diet Intervention on Aldosterone and Levels of mRNA Expression of Urinary Podocin

Figure 18 shows aldosterone concentration in the normal control (NC), prediabetic control (PD), metformin and diet intervention (MET + DI), metformin and high-fat/high-carbohydrate diet (MET + HFHC), rhenium (V) compound and diet intervention (Re + DI) and rhenium (V) compound and high-fat/high-carbohydrate diet (Re + HFHC) groups after 12 weeks of treatment. In comparison with the NC group, the PD group showed a significant increase in aldosterone concentration (*p* < 0.05) (Figure 18). The administration of the rhenium (V) compound in both the diet intervention and high-fat/high-carbohydrate treatments resulted in a significant decrease in aldosterone concentration when compared with the PD group (*p* < 0.05). A similar effect was observed with the MET + DI-treated group when compared with the PD group (*p* < 0.05). See Figure 18.

Figure 19 shows the levels of the mRNA expression of urinary podocin in the normal control (NC), prediabetic control (PD), metformin and diet intervention (MET + DI), metformin and high-fat/high-carbohydrate diet (MET + HFHC), rhenium (V) compound and diet intervention (Re + DI) and rhenium (V) compound and high-fat/high-carbohydrate diet (Re + HFHC) groups after 12 weeks of treatment. In comparison with the NC group, the PD group was shown to have high levels of podocin mRNA expression (*p* < 0.05) (Figure 19). The administration of the rhenium (V) compound in both the diet intervention and high-fat/high-carbohydrate treatments resulted in significantly reduced levels of podocin mRNA when compared with the PD group (*p* < 0.05). A similar effect was observed with the MET + DI-treated group when compared with the PD group (*p* < 0.05). See Figure 19.

## 3. Discussion

There has been growing interest in the use of metal complexes to treat prediabetes in the pharmacotherapy industry [24,25]. The focus of the current study is on a [3+1] oxo-free rhenium (V) compound with uracil-derived ligands in the treatment of prediabetes in diet-induced prediabetic rats.

Type 2 diabetes (T2DM) is a progressive disease that often begins with prediabetes and is characterised by chronic hyperglycaemia caused by insulin resistance [26]. Insulin resistance is a key feature of T2DM and is associated with the cardiorenal metabolic syndrome, leading to the initial stages of cardiovascular and renal diseases in T2DM patients [19,27]. Untreated prediabetic rats had hyperglycaemia and insulin resistance, as shown by the HOMA2-IR index, and also abnormalities in electrolytes and signs of kidney damage, as determined by KIM 1 and low GFR. The groups treated with the rhenium (V) compound in both the presence and absence of a diet intervention had improved glycaemic control, as evidenced by lower glucose levels, low plasma insulin and regulated HOMA-IR values. Only the metformin-treated group with the diet intervention showed a reduction in glucose, insulin and the HOMA-IR value, while the metformin-treated group that consumed an HFHC diet did not show changes in these parameters. This agrees with the literature trend that suggests that metformin is effective with dietary intervention. However, uncontrolled hyperglycaemia and hyperinsulinaemia affect kidney cells during the prediabetic state.

In this study, we looked at hyperglycaemia-induced oxidative stress as a cause of kidney damage. Hyperglycaemia causes the auto-oxidation of glucose, the glycation of proteins and the activation of the polyol mechanism [12,28]. The polyol pathway is activated by hyperglycaemia, especially in non-insulin target tissues, including the kidneys. In this pathway, the conversion of glucose to sorbitol via aldose reduction is at the expense of the overconsumption of NADPH, which is essential for glutathione synthesis, a major antioxidant [29]. The over-production of intracellular reactive oxygen species and antioxidant deficiency contribute to several microvascular and macrovascular complications of the kidney [28,30]. Oxidative stress stimulates the generation of inflammatory mediators and inflammation, which in turn enhance the production of reactive oxygen species [19]. The untreated prediabetic group showed increased levels of the lipid peroxidation marker (MDA) and increased KIM 1 urinary and podocin levels, accompanied by a decline in GFR. Studies on prediabetes and T2DM have shown evidence that the consumption of a high-fat/high-carbohydrate diet has a positive correlation with systemic oxidative stress and renal disease [31,32]. Studies have also reported on the downregulation of the Nrf2/ARE pathway, which is responsible for the expression of antioxidants and anti-inflammatory proteins [33]. Emerging studies have demonstrated that chronic oxidative stress, as occurs in diabetes, leads to the irregular inhibition of the Nrf2/ARE pathway by Keap1 [34]. Recent reports have also suggested that the activation of the Nrf2/ARE pathway can prevent kidney disease progression [35]. Various molecules, such as resveratrol, have been shown to stimulate this pathway, thus ameliorating oxidative stress (Eun Nim Kim). The antioxidative effects observed in this study could partly be attributed to the interaction of the investigated compound with kinases upstream of the NrF2/ARE pathway; however, confirmatory studies are necessary. Furthermore, it has been reported that the upregulation of NAD(P)H oxidase 4 (NOX4) plays an important role in causing renal oxidative stress and kidney injury in animal models of chronic kidney disease (CKD) and diabetic nephropathy (DN) [16,36]. Oxidative stress also plays an important role in podocyte injury and the downregulation of glomerular filtration barrier proteins, nephron and podocin [37,38]. Damage to podocytes results in proteinuria and, eventually, renal failure, which is due to a decrease in glomerular permeability due to mesangial expansion [38,39]. Apart from oxidative stress and associated inflammation, podocyte injury or death could be attributed to insulin resistance [40]. These cells, which are crucial for the integrity of the glomerular basement membrane, have been reported to be insulin-responsive, and hence, their survival is insulin-dependent. Amongst other renal cells, podocytes highly express insulin receptors and Akt2 and GLUT 4 [41,42]. Insulin signalling has been reported to be key in podocyte functions, including the maintenance of glomerulus integrity.

Upon treatment with the rhenium (V) compound, there was improved kidney function. This is observed through the improved GFR, as there is less damage to the kidney, which is evidenced by reduced KIM 1 and podocin levels in rhenium-treated groups. We speculate that the rhenium (V) compound facilitates kidney recovery by first preventing hyperglycaemia-induced oxidative stress by normalizing oxidative stress and antioxidant defence enzymes, as the results show improved antioxidant enzyme activities of SOD and GPx and the suppression of lipid peroxidation in treated prediabetic rats. The ability of the rhenium (V) compound to attenuate hyperglycaemia could also be beneficial in improving kidney function. Many reports agree that adequate glycaemic control can prevent kidney dysfunction or injury. Additionally, the renoprotective effect could also be attributed to the ability of rhenium compounds to enhance the insulin sensitivity of podocytes and, ultimately, their function and survival. A previous study has shown that the administration of this compound led to lower levels of glycated haemoglobin (HbA1c) and increased insulin sensitivity in prediabetic rats [18,43]. The prevention of renal damage in rhenium (V) compound-treated rats in both the presence and absence of a dietary intervention could then explain the significantly lower levels of urine albumin in comparison with untreated prediabetic rats. It is noteworthy that while there are significant improvements in the rhenium (V) compound-treated groups without the dietary intervention, these improvements are not as pronounced as those observed in the groups treated with the dietary intervention. This may be attributable to the continued assault on the kidneys as a result of the continued ingestion of the HFHC diet. In a previous paper, it was shown that even though there is an amelioration of hyperinsulinaemia in the absence of the dietary intervention, the decrease in insulin levels is not as pronounced in comparison with the groups with the dietary intervention [18]. Continued ingestion may still lead to pronounced insulin resistance, which could mask the effect of the rhenium compound. As there are no statistically significant differences between the two groups, this may indicate that there may be improved efficacy with the dietary intervention.

Inflammatory cytokines, including transforming growth factor, are involved in the development and progression of diabetic nephropathy by promoting mesangium expansion and cellular matrix thickening [19,44]. In this study, we looked at IL-6 and TNF-α to check the effects of the rhenium (V) compound on preventing inflammation. Increased levels of IL-6 and TNF-α in the untreated prediabetic group indicated glomerular basement membrane thickening and high levels of IL-6, which are associated with an elevation in its urinary excretion [44]. TNF-α is cytotoxic to renal cells and is able to induce direct renal injury [44,45]. TNF-α also directly induces reactive oxygen species (ROS) in diverse cells, including mesangial cells [19,45]. However, the prediabetic group treated with the rhenium (V) compound had reduced levels of inflammatory markers, and this may be due to the beneficial effects mentioned above, which resulted in the prevention of immune system activation; therefore, the rhenium (V) compound may have anti-inflammatory properties.

Both the inflammation and oxidative stress mentioned above can cause the activation of various pathways, including NF-kB, which further activate apoptosis and cell proliferation and differentiation and lead to the loss of renal function [46,47]. However, NF-Kb can be downregulated by pharmaceutical therapies that have anti-inflammatory and antioxidant properties via targeting the Nrf2 pathway, which has a central role in encoding antioxidants [35,47]. Nrf2 inhibits reactive oxygen species (ROS) and inflammatory pathways that lead to kidney dysfunction [35,48]. We speculate that the rhenium (V) compound targets Keap1, which results in the release of Nrf2 into the nucleus. Once Nrf2 is activated and in the nucleus, it binds to the gene regulator antioxidant response element (ARE) region and mediates the transcription of antioxidant genes. Therefore, antioxidant activity is increased via this pathway, which is supported by the increased activity of SOD and GPx in the rhenium-treated groups.

Prediabetes has been shown to be an intermediate-stage complication associated with T2DM; for example, this is when renal dysfunction often begins [4,49]. Kidney dysfunction is known to be caused by a combination of fat accumulation as a result of insulin resistance in peripheral tissues and primary hypertension [50,51]. In an insulin-resistant state, Na+/K+-ATPase activity is decreased, and insulin resistance induces hyperinsulinaemia [52,53]. Sodium reabsorption from renal tubules is increased and leads to high blood pressure, and lastly, the circulatory fluid volume can also increase relative to hyperglycaemia-induced hyperosmolarity [51,53]. Indeed, rats treated with the rhenium (V) compound in both the presence and absence of the dietary intervention had significantly lower blood pressure as compared to untreated prediabetic rats. We speculate that this could be due to the glucose-metabolism-ameliorating effects of the rhenium (V) compound, as previously shown by Siboto and colleagues [18]. In that study, the rhenium (V) compound reduced hyperinsulinaemia, and this is a positive effect because, when insulin is regulated, Na+/K+-ATPase activity will increase. This leads to sodium not being reabsorbed at a high rate, which results in reduced blood pressure. In metformin-treated rats, the group that received diet modification sustained glomerular capillary hypertension and subsequent glomerular barrier injury and microalbumin leakage; metformin exerts pleiotropic actions on the kidney, beyond its effects as a glucose-lowering agent by attenuating DN, associated with its ability to improve insulin resistance, lipid metabolism, and antioxidative and anti-inflammatory functions [15,54].

Blood pressure is not the only marker that the rhenium (V) compound ameliorated in the treated prediabetic rats; the GFR function was also improved, as we observed that the treated group had improved electrolyte reabsorption and excretion. Studies have shown that diabetic nephropathy is associated with a decline in renal function, including the reduced function of the glomerular filtration rate (GFR), which is associated with electrolyte abnormalities, water imbalance and hypernatremia [51,55]. Elevated plasma uric acid is associated with a higher risk of insulin resistance [56,57]. Uric acid is a weak acid generated by purine metabolism. It has been recognised as the cause of gout since the early 1800s [56,57]. High circulating uric acid levels might increase the risk of T2DM and metabolic syndrome, thereby contributing to a higher risk of diabetic complications among T2DM patients (26). Experiments in rats showed an association between lower serum uric acid levels and improved insulin sensitivity [43].

Diet-induced prediabetic rats had high uric acid levels in comparison with non-prediabetic rats. However, rats treated with the rhenium (V) compound in both the presence and absence of the diet intervention had reduced uric acid, suggesting the amelioration of uric acid. Lastly, the significantly increased urea, creatinine and uric acid levels in untreated prediabetic rats demonstrated renal damage and metabolic alterations resulting from insulin resistance and hyperglycaemia. However, treatment with the rhenium (V) compound in both the presence and absence of the dietary intervention resulted in a significant decrease in creatinine, plasma urea and uric acid levels, signifying its nephroprotective potential.

Prediabetes is associated with the increased activity of the renin–angiotensin–aldosterone system (RAAS), resulting in the retention of sodium and water and a rise in blood pressure [14,53]. The upregulation of RAAS contributes to the impairment of the renal system, which was evidenced by the significantly increased concentration of aldosterone in the untreated prediabetic group [32,58]. High levels of aldosterone have negative effects on the system, such as inflammation, oxidative stress and even insulin resistance [32,59]. Arterial hypertension plays an important role in the development of kidney dysfunction by causing glomerular hyperfiltration and glomerular hypertrophy, followed by the expansion of the mesangium and the accumulation of the extracellular matrix [55,60]. Prediabetic rats treated with the rhenium (V) compound in both the presence and absence of the dietary intervention had lower levels of aldosterone in comparison with untreated prediabetic rats. Metal complexes including transition metals such as vanadium have been shown to be useful in treating prediabetic hypertension by targeting the RAAS by decreasing angiotensin concentrations [61]. We speculate that the rhenium (V) compound also uses the same mechanism to reduce aldosterone and high blood pressure, but further studies will be conducted on the effects of the rhenium (V) compound on regulating blood pressure, including mean arterial pressure [61].

## 4. Materials and Methods

### 4.1. Animals

Thirty-six [36] male Sprague-Dawley rats (150–180 g) obtained from Biomedical Research Unit, University of KwaZulu-Natal (UKZN), were kept under standard environmental conditions, i.e., constant humidity (55 ± 5%), temperature (22 ± 2 °C) and 12 h day/12 h night cycle. The animals were acclimatised for 2 weeks and consumed standard rat chow (Meadow Feeds, South Africa) and water ad libitum before being fed on the experimental high-fat/high-carbohydrate (HFHC) diet (AVI Products (Pty) Ltd., Waterfall, South Africa) to induce prediabetes. The HFHC diet consists of carbohydrates (55% kcal/g), fats (30% kcal/g) and proteins (15% kcal/g), as described in our previous studies [18,31]. All of the experimental designs and procedures were carried out according to the ethics and guidelines of the Animal Research Ethics Committee (AREC, ethical clearance code: AREC/00003221/2021) of UKZN, Durban, South Africa.

### 4.2. Induction of Prediabetes

Sprague-Dawley rats were randomly assigned to the following diet groups (n = 6 per group): a standard rat chow with normal drinking water (ND + H_2_O) or a high-fat/high-carbohydrate diet with drinking water supplemented with fructose (HFHC + Fructose). The experimental prediabetes induction period was 20 weeks [31,62]. Rats with fasting blood glucose of more than 5.6 mmol/L were considered prediabetic and further grouped for pharmacological studies [63,64]. The treatment started on the subsequent day, and this was considered the first day of treatment.

### 4.3. Experimental Design

The animals were randomly divided into 6 groups of 6 animals in each (30 with persisting prediabetes; 6 normal). Group 1: Normal healthy control rats received vehicle (NC); Group 2: Prediabetic control rats continued with the HFHC diet and received vehicle (PD); Group 3: Treated prediabetic rats continued with the STD diet and received metformin (MET + DI); Group 4: Treated prediabetic rats continued with the HFHC diet and received metformin (MET + HFHC); Group 5: Prediabetic rats continued with the STD diet and received the rhenium (V) compound (Re + DI); Group 6: Treated prediabetic rats continued with the HFHC diet and received the rhenium (V) compound (Re + HFHC).

### 4.4. Treatment of Prediabetic Animals

The treatment period started after 20 weeks of prediabetes induction and lasted an additional 12 weeks. The animals were treated once every third day at 9:00 a.m., where the MET + HFHC and MET + DI groups received an oral dose of metformin (500 mg/kg), while the Re + HFHC and Re + DI groups received a subcutaneous injection of the rhenium (V) compound (15 mg/kg). The concentration of the rhenium (V) compound was chosen from a previous study conducted in our laboratory [18]. Parameters including blood glucose, blood pressure, fluid intake and urine output were monitored every 4 weeks during the treatment period.

#### 4.4.1. Determination of Fluid Intake and Urine Output

At the beginning of the treatment period and every 4 weeks thereafter, all of the animals in each group were placed in different metabolic cages for 24 h to measure fluid intake and urine output. The urine samples were measured and centrifuged at 13,000 rpm for 5 min at 4 °C, and the supernatants were stored at −80 °C in a Bio Ultra freezer (Snijders Scientific, Tilburg, Holland) until ready for kidney function parameter analysis.

#### 4.4.2. Blood Collection and Tissue Harvesting

For blood collection, all animals were anaesthetised with Isofor (100 mg/kg) (Safeline Pharmaceuticals (Pty) Ltd., Roodeport, South Africa), 4–5% for the induction and 1–2% for maintenance, as is recommended by the anaesthesia guideline, and Isofor showed no negative health effects on the rats when administered via a gas anaesthetic chamber (Biomedical Resource Unit, UKZN, Durban, South Africa) for 3 min. While rats were unconscious, blood was collected by cardiac puncture into individual pre-cooled heparinised containers. The blood was then centrifuged (Eppendorf centrifuge 5403, Hamburg, Germany) at 4 °C and 503 g for 15 min. Plasma was collected and stored at −80 °C in a Bio Ultra freezer (Snijers Scientific labs, Holland, The Netherlands) until ready for biochemical analysis. Thereafter, kidney tissue was removed, weighed and rinsed with cold normal saline solution and snap frozen in liquid nitrogen before storage in a BioUltra freezer (Snijers Scientific, Tilburg, Netherlands) at −80 °C until biochemical analysis.

### 4.5. Biochemical Analysis

The biochemical analysis of kidney function parameters (such as creatinine, urea, uric acid, albumin and electrolytes (Na^+^ and K^+^)) was performed on plasma and urine samples in the 32nd week by using their respective assay kits (Elabscience Biotechnology Co., Ltd., Houston, TX, USA) as instructed by the manufacturer. However, the kidney injury molecule (KIM-1) and aldosterone plasma concentrations were determined using their specific ELISA kits (KIM 1: E-EL-R3019; aldosterone: E-EL-0070) as instructed by the manufacturer (Elabscience Biotechnology Co., Ltd., Houston, TX, USA) via a microplate reader (SPECTROstar Nano spectrophotometer; BMG LABTECH, Ortenburg, LGBW, Germany).

#### 4.5.1. Determination of GFR

The GFRs of all animals were determined in the 32nd week of the experiment from the estimation of creatinine in the plasma and urine (creatinine clearance) as follows: GFR mL/min = Urine creatinine mg/dL × 24 h urine volume mL
Plasma creatinine mg/dL × 60 min × 24 h

#### 4.5.2. Lipid Peroxidation and Antioxidant Status

Lipid peroxidation was assessed by determining the concentration of malondialdehyde (MDA) in homogenised kidney tissues according to a previously established protocol [65]. However, the antioxidant status of the kidney homogenate was assessed by determining the concentration of superoxide dismutase (SOD) and glutathione peroxidase (GPx) by using their specific ELISA kits (SOD: E-EL-R1324; GPx: E-EL-R2491) according to the instructions of the manufacturer (Elabscience Biotechnology Co., Ltd., Houston, TX, USA). The inflammatory markers TNF-α and IL-6 were measured using their specific ELISA kits (Rat TNF-α: E-EL-R2856; Rat IL-6: E-EL-R0015).

#### 4.5.3. Urine RNA Isolation

The kit was purchased from Inqaba Biotechnical Industries (pty) Ltd. RNA was isolated from urine (4 mL) by using the ZR Urine RNA Isolation Kit™ (Zymo Research Corp., Irvine, CA, USA) according to the manufacturer’s protocol. RNA was treated with DNAse before reverse transcription. The purity of the RNA was confirmed by the relative absorbance ratio of 260/280 nm via a Nanodrop 1000 spectrophotometer (Thermo Scientific, Santa Clara, CA, USA).

#### 4.5.4. Urine Complementary DNA (cDNA) Synthesis

Urine RNA (100 ng) was reverse-transcribed to complementary DNA (cDNA) by using the iScript™ cDNA Synthesis Kit (Bio-Rad, Hercules, CA, USA) through incubation in a thermal cycler (SimpliAmp Thermal Cycler, Applied Biosystems, Life Technologies, thermo fisher scientific, Johannesburg, South Africa). For cDNA synthesis, urine RNA (2 μL) was mixed with 5× iScript reaction (4 μL), iScript reverse transcriptase enzyme (1 μL) (Bio-Rad, Hercules, CA, USA) and nuclease-free water to a final volume of 20 μL. The mixture was incubated in the thermal cycler (SimpliAmp Thermal Cycler, Applied Biosystems, Life Technologies; thermo fisher scientific, Johannesburg, South Africa) at 25 °C for 5 min, 42 °C for 30 min and, finally, 85 °C for 5 min. Thereafter, the synthesised cDNA was stored at −80 °C until its use for real-time PCR (polymerase chain reaction).

#### 4.5.5. Real-Time PCR

The urinary mRNA level of podocin was quantified by a real-time PCR LightCycler (Roche LightCycler 96, Mississauga, ON, Canada). cDNA template (2 μL), SYBR Green PCR master mix (5 μL) (Bio-Rad, Hercules, CA, USA), podocin forward primer (1 μL), podocin reverse primer (1 μL) and nuclease-free water were mixed to a final volume of 10 μL. Thereafter, the sample mixtures were cycled 40 times at 95 °C for 10 s, 60 °C for 20 s, and 72 °C for 20 s in the LightCycler (Roche LightCycler 96, Mississauga, ON, Canada). All samples were run in duplicate, and β-actin mRNA was used as a housekeeping gene (β-actin primers: forward primer GCA CCA CAC CTT CTA CAA TG; reverse primer TGC TTG CTG ATC CAC ATC TG) to normalise the podocin mRNA level. The sequences of the used oligonucleotide primers (Metabion International AG, Planegg, Germany) were as follows: podocin forward 5′-TGG AAG CTG AGG CAC AAA GA-3′ and podocin reverse 5′-AGA ATC TCA GCC GCC ATC CT-3′. 

### 4.6. Statistical Analysis

All data are expressed as means ± SEM. Statistical comparisons were performed with Graph Pad In Stat Software (version 5.00, Graph Pad Software, Inc., San Diego, CA, USA) using one-way analysis of variance (ANOVA) followed by the Tukey–Kramer multiple comparison test. A value of *p* < 0.05 was considered statistically significant

## 5. Conclusions

The treatment of diet-induced prediabetic rats with the rhenium (V) compound with uracil-derived ligands in both the presence and absence of a diet intervention did not only markedly improve insulin sensitivity but also effectively decreased metabolic disturbances, thereby preventing hyperglycaemia-induced oxidative stress in the kidney. We believe that these findings warrant more studies into the biological activity of this compound, as it may have the potential to not only restore glucose homeostasis in the prediabetic state but also prevent prediabetes-associated complications.

## Figures and Tables

**Figure 1 ijms-23-15400-f001:**
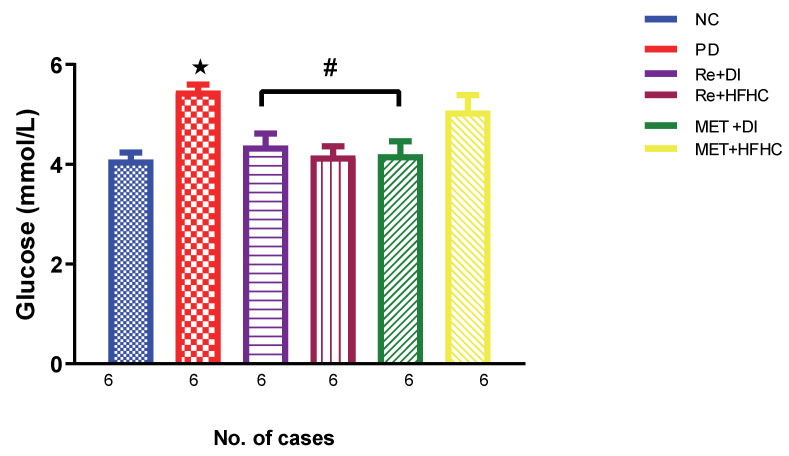
Glucose levels in normal control (NC), prediabetic control (PD), metformin and diet intervention (MET + DI), metformin and high-fat/high-carbohydrate diet (MET + HFHC), rhenium (V) compound and diet intervention (Re + DI) and rhenium (V) compound and high-fat/high-carbohydrate diet (Re + HFHC) groups after 12 weeks of treatment. Values are presented as means ± SEM (n = 6). ★ *p* < 0.05 in comparison with NC; # *p* < 0.05 in comparison with PD, MET + DI, MET + HFHC, Re + DI and Re + HFHC.

**Figure 2 ijms-23-15400-f002:**
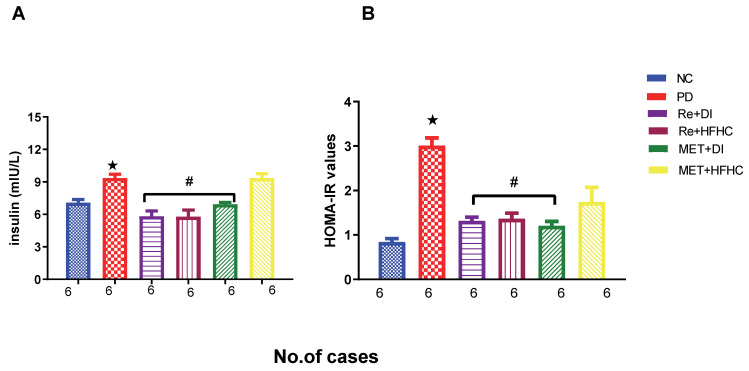
(**A**,**B**) Insulin concentration and HOMA2-IR index in normal control (NC), prediabetic control (PD), metformin and diet intervention (MET + DI), metformin and high-fat/high-carbohydrate diet (MET + HFHC), rhenium (V) compound and diet intervention (Re + DI) and rhenium (V) compound and high-fat/high-carbohydrate diet (Re + HFHC) groups after 12 weeks of treatment. Values are presented as means ± SEM (n = 6). ★ *p* < 0.05 in comparison with NC; # *p* < 0.05 in comparison with PD, MET + DI, MET + HFHC, Re + DI and Re + HFHC.

**Figure 3 ijms-23-15400-f003:**
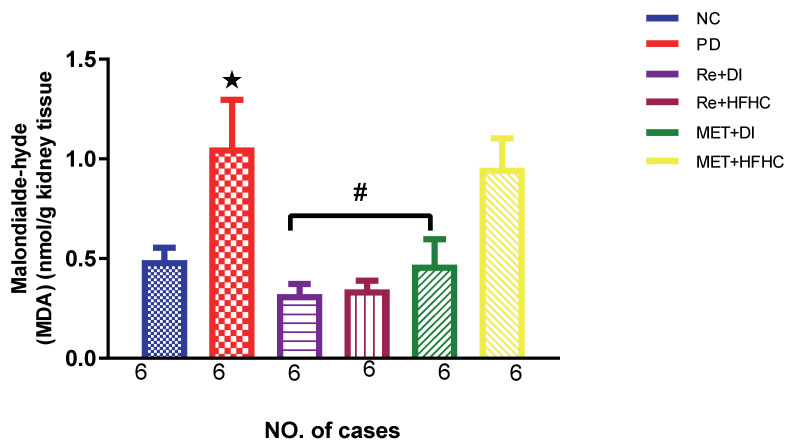
MDA concentration in normal control (NC), prediabetic control (PD), metformin and diet intervention (MET + DI), metformin and high-fat/high-carbohydrate diet (MET + HFHC), rhenium (V) compound and diet intervention (Re + DI), and rhenium (V) compound and high-fat/high-carbohydrate diet (Re + HFHC) groups after 12 weeks of treatment. Values are presented as means ± SEM (n = 6). ★ *p* < 0.05 in comparison with NC; # *p* < 0.05 in comparison with PD, MET +DI, MET + HFHC, Re + DI and Re + HFHC.

**Figure 4 ijms-23-15400-f004:**
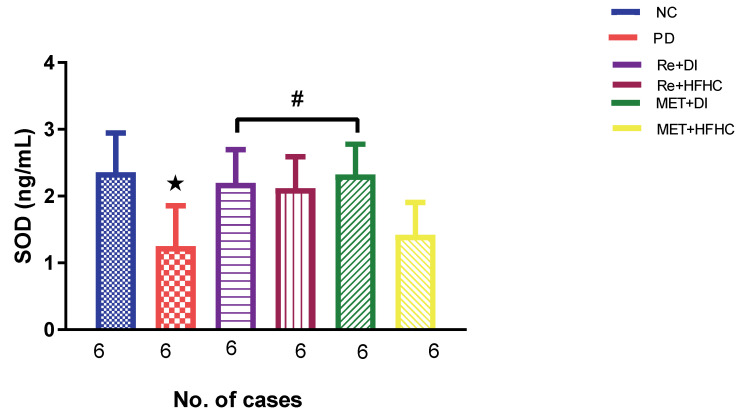
SOD activity in normal control (NC), prediabetic control (PD), metformin and diet intervention (MET + DI), metformin and high-fat/high-carbohydrate diet (MET + HFHC), rhenium (V) compound and diet intervention (Re + DI), and rhenium (V) compound and high-fat/high-carbohydrate diet (Re + HFHC) groups after 12 weeks of treatment. Values are presented as means ± SEM (n = 6). ★ *p* < 0.05 in comparison with NC; # *p* < 0.05 in comparison with PD, MET +DI, MET + HFHC, Re + DI and Re + HFHC.

**Figure 5 ijms-23-15400-f005:**
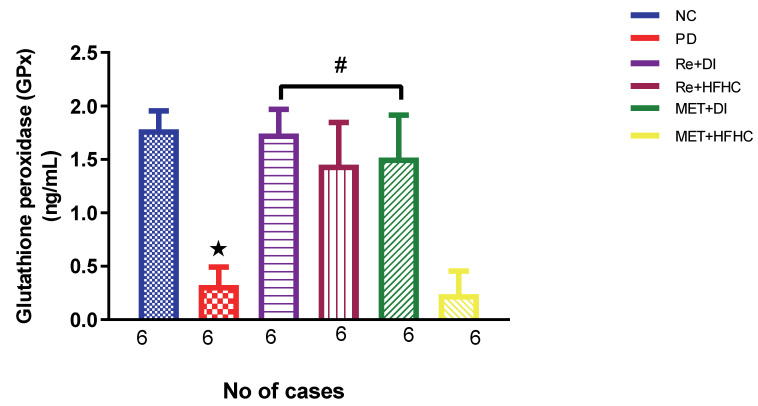
Shows GPx activity in normal control (NC), prediabetic control (PD), metformin and diet intervention (MET + DI), metformin and high-fat/high-carbohydrate diet (MET + HFHC), rhenium (V) compound and diet intervention (Re + DI), and rhenium (V) compound and high-fat/high-carbohydrate diet (Re + HFHC) groups after 12 weeks of treatment. Values are presented as means ± SEM (n = 6). ★ *p* < 0.05 in comparison with NC; # *p* < 0.05 in comparison with PD, MET +DI, MET + HFHC, Re + DI and Re + HFHC.

**Figure 6 ijms-23-15400-f006:**
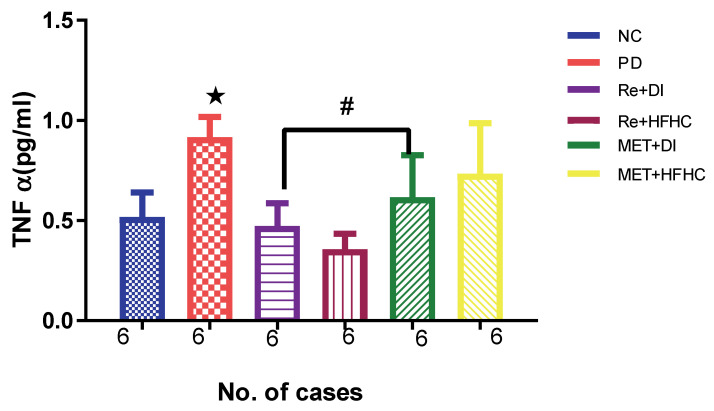
TNF α in normal control (NC), prediabetic control (PD), metformin and diet intervention (MET + DI), metformin and high-fat/high-carbohydrate diet (MET + HFHC), rhenium (V) com/pound and diet intervention (Re + DI), and rhenium (V) compound and high-fat/high-carbohydrate diet (Re + HFHC) groups after 12 weeks of treatment. Values are presented as means ± SEM (n = 6). ★ *p* < 0.05 in comparison with NC; # *p* < 0.05 in comparison with PD, MET + DI, MET + HFHC, Re + DI and Re + HFHC.

**Figure 7 ijms-23-15400-f007:**
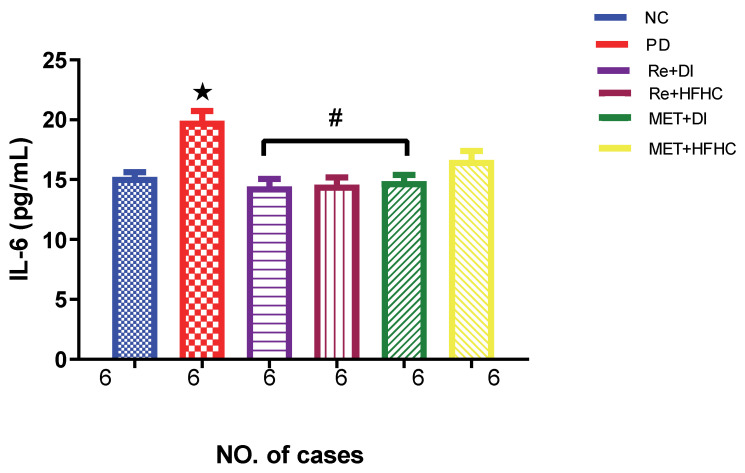
IL-6 concentration in normal control (NC), prediabetic control (PD), metformin and diet intervention (MET + DI), metformin and high-fat/high-carbohydrate diet (MET + HFHC), rhenium (V) compound and diet intervention (Re + DI), and rhenium (V) compound and high-fat/high-carbohydrate diet (Re + HFHC) groups after 12 weeks of treatment. Values are presented as means ± SEM (n = 6). ★ *p* < 0.05 in comparison with NC; # *p* < 0.05 in comparison with PD, MET + DI, MET + HFHC, Re + DI and Re + HFHC.

**Figure 8 ijms-23-15400-f008:**
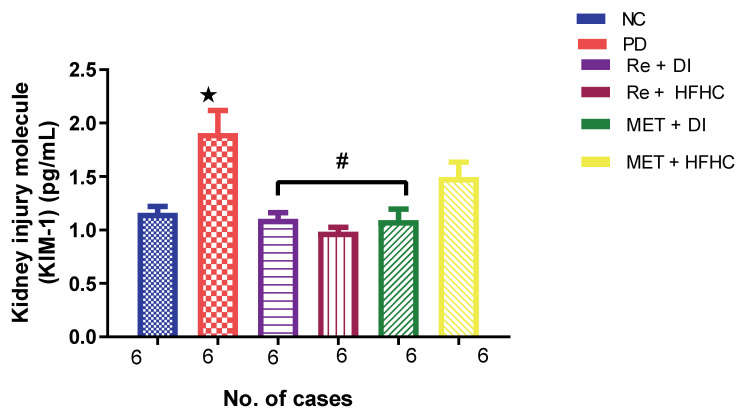
Plasma KIM-1 concentration in normal control (NC), prediabetic control (PD), metformin and diet intervention (MET + DI), metformin and high-fat/high-carbohydrate diet (MET + HFHC), rhenium (V) compound and diet intervention (Re + DI) and rhenium (V) compound and high-fat/high-carbohydrate diet (Re + HFHC) groups after 12 weeks of treatment. Values are presented as means ± SEM (n = 6). ★ *p* < 0.05 in comparison with NC; α *p* < 0.05 in comparison with PC, MET + DI, MET + HFHC, Re + DI and Re + HFHC. ★ *p* < 0.05 in comparison with NC; # *p* < 0.05 in comparison with PD, MET + DI, MET + HFHC, Re + DI and Re + HFHC.

**Figure 9 ijms-23-15400-f009:**
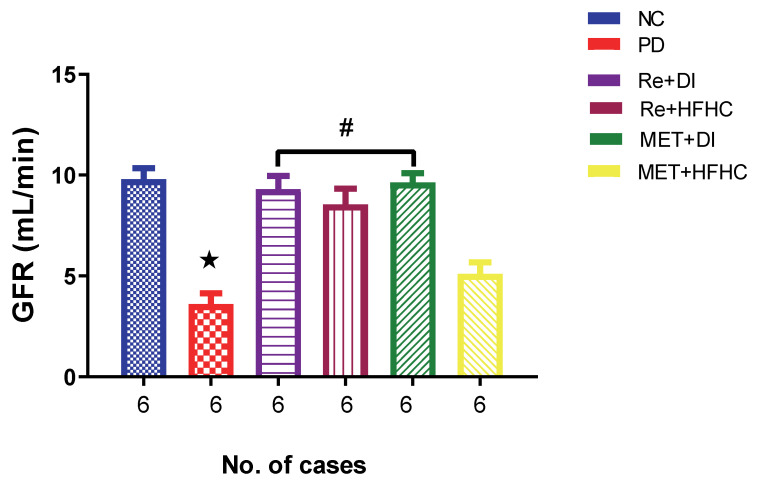
GFR in normal control (NC), prediabetic control (PD), metformin and diet intervention (MET + DI), metformin and high-fat/high-carbohydrate diet (MET + HFHC), rhenium (V) compound and diet intervention (Re + DI) and rhenium (V) compound and high-fat/high-carbohydrate diet (Re + HFHC) groups after 12 weeks of treatment. Values are presented as means ± SEM (n = 6). ★ *p* < 0.05 in comparison with NC; # *p* < 0.05 in comparison with PC, MET + DI, MET + HFHC, Re + DI and Re + HFHC.

**Figure 10 ijms-23-15400-f010:**
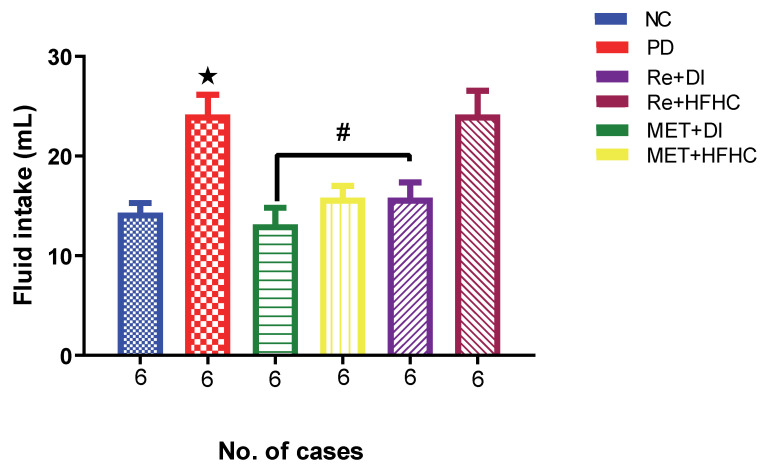
Fluid intake volume in normal control (NC), prediabetic control (PD), metformin and diet intervention (MET + DI), metformin and high-fat/high-carbohydrate diet (MET + HFHC), rhenium (V) compound and diet intervention (Re + DI) and rhenium (V) compound and high-fat/high-carbohydrate diet (Re + HFHC) groups after 12 weeks of treatment. Values are presented as means ± SEM (n = 6). ★ *p* < 0.05 in comparison with NC; # *p* < 0.05 in comparison with PC, MET + DI, MET + HFHC, Re + DI and Re + HFHC.

**Figure 11 ijms-23-15400-f011:**
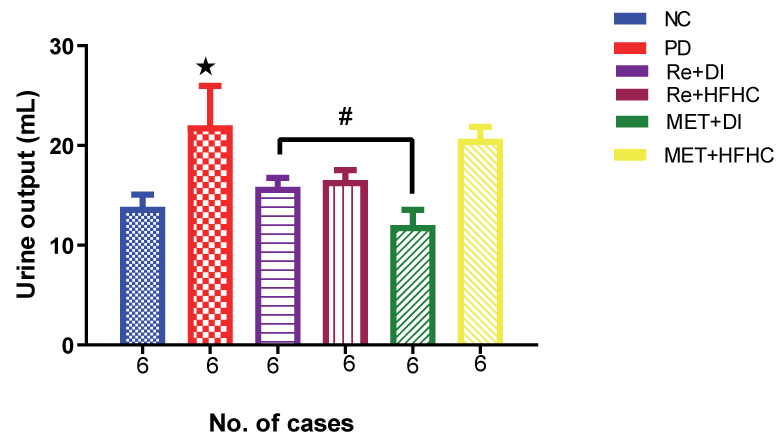
Urinary output volume in normal control (NC), prediabetic control (PD), metformin and diet intervention (MET + DI), metformin and high-fat/high-carbohydrate diet (MET + HFHC), rhenium (V) compound and diet intervention (Re + DI) and rhenium (V) compound and high-fat/high-carbohydrate diet (Re + HFHC) groups after 12 weeks of treatment. Values are presented as means ± SEM (n = 6). ★ *p* < 0.05 in comparison with NC; # *p* < 0.05 in comparison with PC, MET + DI, MET + HFHC, Re + DI and Re + HFHC.

**Figure 12 ijms-23-15400-f012:**
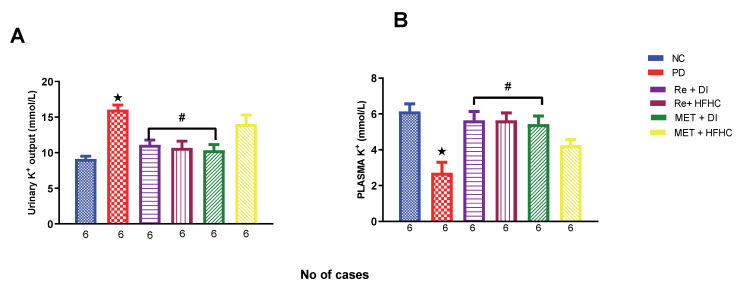
(**A**,**B**) Urinary and plasma K^+^ concentration in normal control (NC), prediabetic control (PD), metformin and diet intervention (MET + DI), metformin and high-fat/high-carbohydrate diet (MET + HFHC), rhenium (V) compound and diet intervention (Re + DI) and rhenium (V) compound and high-fat/high-carbohydrate diet (Re + HFHC) groups after 12 weeks of treatment. Values are presented as means ± SEM (n = 6). ★ *p* < 0.05 in comparison with NC; # *p* < 0.05 in comparison with PC, MET + DI, MET + HFHC, Re + DI and Re + HFHC.

**Figure 13 ijms-23-15400-f013:**
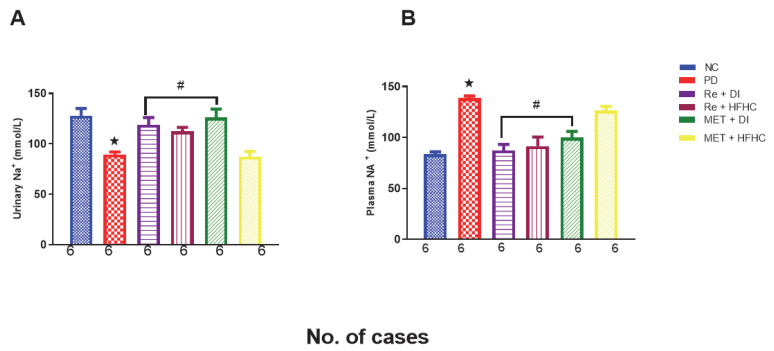
(**A**,**B**) Urinary and plasma NA^+^ concentration in normal control (NC), prediabetic control (PD), metformin and diet intervention (MET + DI), metformin and high-fat/high-carbohydrate diet (MET + HFHC), rhenium (V) compound and diet intervention (Re + DI) and rhenium (V) compound and high-fat/high-carbohydrate diet (Re + HFHC) groups after 12 weeks of treatment. Values are presented as means ± SEM (n = 6). ★ *p* < 0.05 in comparison with NC; # *p* < 0.05 in comparison with PC, MET + DI, MET + HFHC, Re + DI and Re + HFHC.

**Figure 14 ijms-23-15400-f014:**
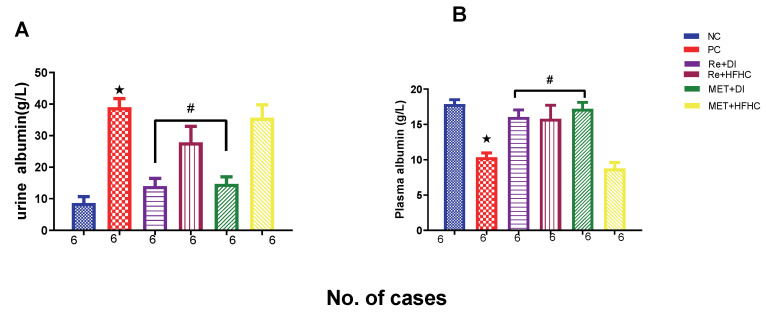
(**A**,**B**) Plasma and urinary albumin concentrations in normal control (NC), prediabetic control (PD), metformin and diet intervention (MET + DI), metformin and high-fat/high-carbohydrate diet (MET + HFHC), rhenium (V) compound and diet intervention (Re + DI) and rhenium (V) compound and high-fat/high-carbohydrate diet (Re + HFHC) groups after 12 weeks of treatment. Values are presented as means ± SEM (n = 6). ★ *p* < 0.05 in comparison with NC; # *p* < 0.05 in comparison with PC, MET + DI, MET + HFHC, Re + DI and Re + HFHC.

**Figure 15 ijms-23-15400-f015:**
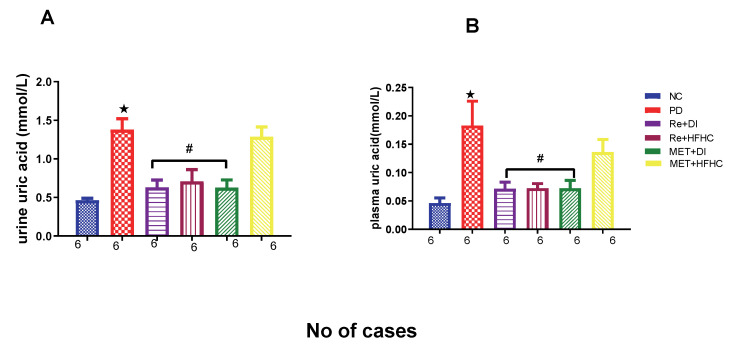
(**A**,**B**) Plasma and urinary uric acid concentrations in normal control (NC), prediabetic control (PD), metformin and diet intervention (MET + DI), metformin and high-fat/high-carbohydrate diet (MET + HFHC), rhenium (V) compound and diet intervention (Re + DI) and rhenium (V) compound and high-fat/high-carbohydrate diet (Re + HFHC) groups after 12 weeks of treatment. Values are presented as means ± SEM (n = 6). ★ *p* < 0.05 in comparison with NC; # *p* < 0.05 in comparison with PC, MET + DI, MET + HFHC, Re + DI and Re + HFHC.

**Figure 16 ijms-23-15400-f016:**
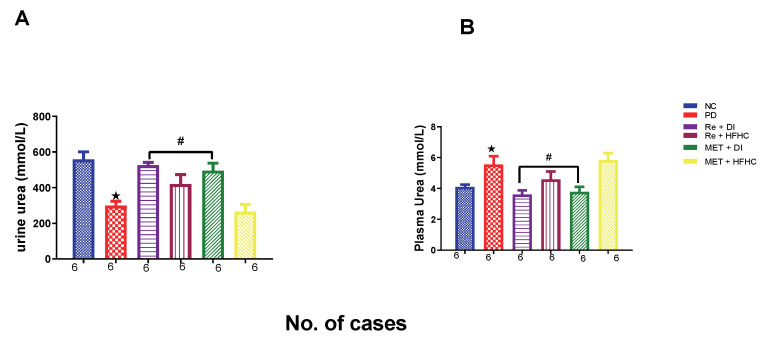
(**A**,**B**) Urinary and plasma urea concentrations in normal control (NC), prediabetic control (PD), metformin and diet intervention (MET + DI), metformin and high-fat/high-carbohydrate diet (MET + HFHC), rhenium (V) compound and diet intervention (Re + DI) and rhenium (V) compound and high-fat/high-carbohydrate diet (Re + HFHC) groups after 12 weeks of treatment. Values are presented as means ± SEM (n = 6). ★ *p* < 0.05 in comparison with NC; # *p* < 0.05 in comparison with PC, MET + DI, MET + HFHC, Re + DI and Re + HFHC.

**Figure 17 ijms-23-15400-f017:**
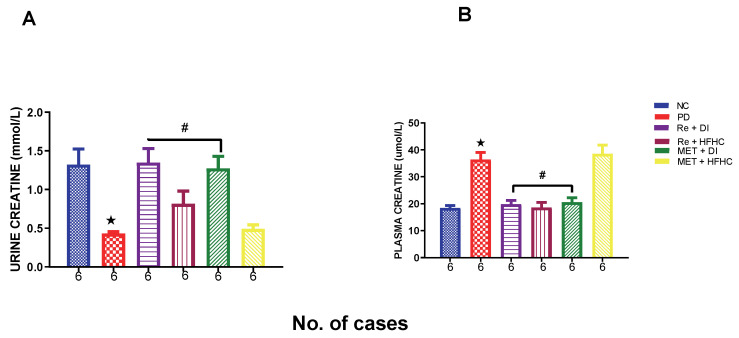
(**A**,**B**) Urinary and plasma creatinine concentrations in normal control (NC), prediabetic control (PD), metformin and diet intervention (MET + DI), metformin and high-fat/high-carbohydrate diet (MET + HFHC), rhenium (V) compound and diet intervention (Re + DI) and rhenium (V) compound and high-fat/high-carbohydrate diet (Re + HFHC) groups after 12 weeks of treatment. Values are presented as means ± SEM (n = 6). ★ *p* < 0.05 in comparison with NC; # *p* < 0.05 in comparison with PC, MET + DI, MET + HFHC, Re + DI and Re + HFHC.

**Figure 18 ijms-23-15400-f018:**
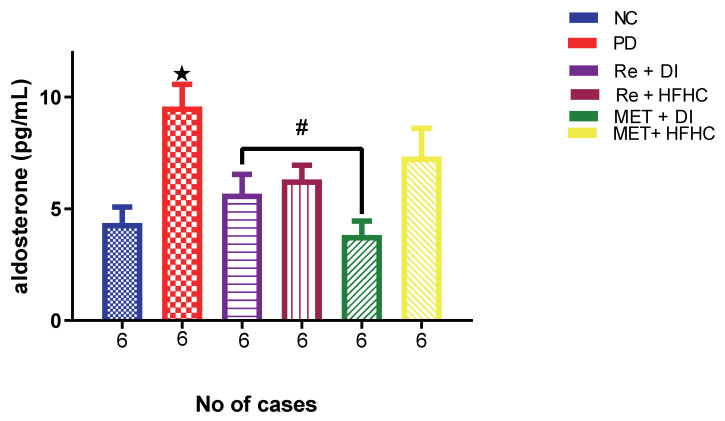
Plasma aldosterone concentration in normal control (NC), prediabetic control (PD), metformin and diet intervention (MET + DI), metformin and high-fat/high-carbohydrate diet (MET + HFHC), rhenium (V) compound and diet intervention (Re + DI) and rhenium (V) compound and high-fat/high-carbohydrate diet (Re + HFHC) groups after 12 weeks of treatment. Values are presented as means ± SEM (n = 6). ★ *p* < 0.05 in comparison with NC; # *p* < 0.05 in comparison with PC, MET + DI, MET + HFHC, Re + DI and Re + HFHC.

**Figure 19 ijms-23-15400-f019:**
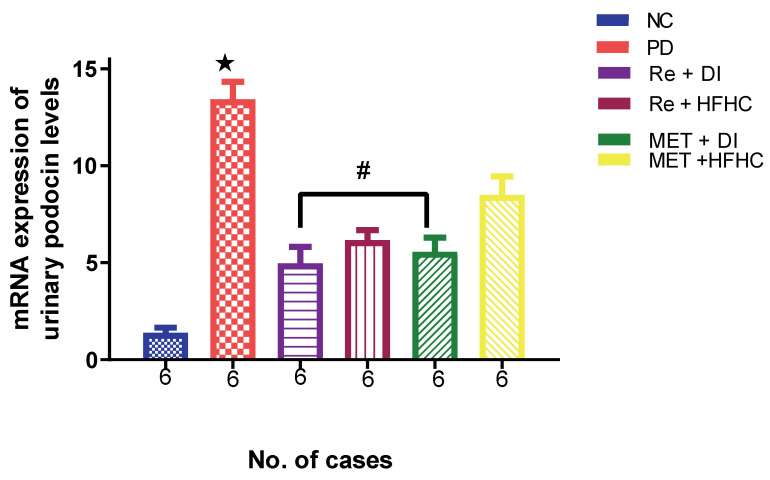
Levels of mRNA expression of urinary podocin in normal control (NC), prediabetic control (PD), metformin and diet intervention (MET + DI), metformin and high-fat/high-carbohydrate diet (MET + HFHC), rhenium (V) compound and diet intervention (Re + DI) and rhenium (V) compound and high-fat/high-carbohydrate (Re + HFHC) groups after 12 weeks of treatment. Values are presented as means ± SEM (n = 6). ★ *p* < 0.05 in comparison with NC; # *p* < 0.05 in comparison with PC, MET + DI, MET + HFHC, Re + DI and Re + HFHC.

## Data Availability

The data presented in this study are available on request from the corresponding author.

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
