# Peer review of "Ameliorative Effects of a Rhenium (V) Compound with Uracil-Derived Ligand Markers Associated with Hyperglycaemia-Induced Renal Dysfunction in Diet-Induced Prediabetic Rats"

_ijms, 2022, doi:10.3390/ijms232315400_

Round 1

Reviewer 1 Report

Manuscript: Rhenium (V) compound with uracil derived ligands amelio rates renal dysfunction by suppressing hyperglycaemia mediated renal oxidative stress and inflammation in diet induced  prediabetic rats

Article Type: Original

In the present article, the authors prove the beneficial effects of the administration rhenium derived (V) compound on kidney damage in prediabetes and high fat-high carbohydrate diet rats, and its administration with dietary control. The authors are able to demonstrate that the rhenium derived (V) compound administration decreases hyperglycaemia, favoring the decrease in the oxidative stress levels in the kidney and aldosterone levels, thus decreasing markers of kidney damage in plasma and urine and electrolyte imbalance. However, the authors do not explore or explain the molecular mechanisms associated with the rhenium (V) protective effects, they do not evaluate the modulation of rhenium (V) in key mechanism for the development of kidney damage like inflammations, apoptosis, fibrosis, metabolic reprograming or antioxidant pathways such as Nrf2, that would be fundamental to explain the observed results. Thus, discussion many times the remains in the descriptive part of the phenomenon. Therefore, the conclusions are excessive, as there are not renal metabolism evaluations in the work, the same thing happens in the title, in which inflammatory processes are mentioned, without enough inflammatory markers evaluation in this article nor is there any histological evidence to support it. Therefore, a deeper study of kidney inflammation or renal metabolic parameters such as mitochondrial bioenergetics, beta oxidation evaluation, glycolytic flux, etc. must be carried out before the article is considered for publication.

Reviewer 2 Report

The authors explored the effects of Rhenium associated with diet intervention in pre-diabetic rats and found that this treatment improved insulin sensitivity and prevented hyperglycaemia-induced oxidative stress.

Comments

1.      All the abbreviations should be disclosed on first use including the Abstract body.

2.      Introduction: The authors should describe previous studied on Rhenium compound and the rationale for its studies in pre-diabetic rats. This should be corrected.

3.      Results: The data from all the Tables should be presented in Graph form. This should be corrected.

4.      All the Figures should indicate the number of cases under each column. This should be corrected.

5.      On all the Figures the authors should indicate two columns by the line over them where the differences were significant. This should be corrected.

6.      The data indicated on Figures 5 and 6 do not correspond to the description in the text. This should be corrected.

7.      Lines 283-294 repeat the information presented in the Introduction. All the repeats should be removed.

8.      Lines 308-314: This information should be moved to Introduction.

9.      Lines 299-307: It is not clear whether this information related to the present study or not and why it is placed here. This should be corrected.

10.  Lines 347-362: This piece is not related to the study. It should be removed or rephrased.

11.  Lines 440-445: The authors should explain why this concentration of Rhenium compound was chosen for his study and toxicity assay of the drug should be presented.

12.  Lines 466-473: The authors should indicate the catalog number of each kit used for ELISA assay.

13.  Lines 490-493 should be moved between lines 494 and 495.

Reviewer 3 Report

Siboto et al studied the effect of Rhenium (V) compound with uracil derived ligands and observed that it reduces renal oxidation and inflammation in diet induced prediabetic rats. They compared the effect of Rhenium with traditional drug metformin that are long time used to control prediabetic. They showed rhenium works better than metformin in high fat diet feeding rats and raised the possibilities of rhenium as a potentially better diabetes drug when patients do not abide by lifestyles changes while taking metformin.

This paper is a good study comparing two drugs in prediabetic rats and should be published. However, there are some questions/concerns that could be answered.

1.       Authors should give a brief description about rhenium V in the introduction (structure, properties, mechanism of action, extracts etc ) so the reader can understand what type of compound it is although it appears that they were working with it long time.

2.       Why the authors can not use rhenium V orally instead of injection? A chemical compound may not be digested by rat intestinal enzyme. Or already they tried it? Because metformin is a oral drug so it will be still more attractable choice than a intravenous drug.  

3.        In figure 3a, rhenium with High fat diet showed similar effect as with metformin and HFHC for urine albumin level. It needs some explanation in the discussion why it is not reduced.

4.        Subsequently, urinary creatine concentration (Fig 6a ) in RE-HFHC group is not elevated as expected from RE-DI group. An explanation could be helpful.

5.       During RNA isolation, whether RNA is treated with DNAse before reverse transcription? Kit name is given but this information is necessary whether the kit contains DNAse step as one of the isolation procedures.

6.       The beta-actin primers or the company name should be given.

Round 2

Reviewer 1 Report

Although the authors do not have time to carry out the necessary experiments to complement the article,  the authors must  to take into account in greater depth in disscussion section, the roles of kidney inflammation, the alterations in renal metabolism and antioxidant pathways like Nrf2, that would be fundamental to explain the observed results.

Reviewer 2 Report

The authors improved their manuscript, however, some previous Comments were not addressed.

Comments

1.      Previous Comment #5 was not addressed. In all the presented graphs it is completely not clear which differences are considered significant. The authors should use only asterisks to compare data. The authors should follow the Reviewer suggestion exactly as it was written previously. This should be done.

2.      Abstract: The authors should indicate in the Abstract what kind of “lifestyle intervention” they address.

3.      All the typos should be corrected.

4.      All the references should be presented according to the requirements of IJMS.

5.      Lines 469-471: This sentence is not clear/ It should be clarified.

Round 3

Reviewer 2 Report

I have no more comments.